# *Acanthaster planci* Inhibits PCSK9 and Lowers Cholesterol Levels in Rats

**DOI:** 10.3390/molecules26165094

**Published:** 2021-08-23

**Authors:** Nurjannatul Naim Kamaruddin, Nor Azwin Hajri, Yosie Andriani, Aina Farahiyah Abdul Manan, Tengku Sifzizul Tengku Muhammad, Habsah Mohamad

**Affiliations:** Institute of Marine Biotechnology, Universiti Malaysia Terengganu, Kuala Nerus 21030, Malaysia; jannatulkamaruddin@yahoo.com (N.N.K.); awin.hajri@gmail.com (N.A.H.); yosie.hs@umt.edu.my (Y.A.); ainafarahiyah91@gmail.com (A.F.A.M.)

**Keywords:** *Acanthaster planci*, PCSK9, atherosclerosis, LDL-receptor, LDL-cholesterol

## Abstract

Atherosclerosis is the main cause of cardiovascular diseases which in turn, lead to the highest number of mortalities globally. This pathophysiological condition is developed due to a constant elevated level of plasma cholesterols. Statin is currently the widely used treatment in reducing the level of cholesterols, however, it may cause adverse side effects. Therefore, there is an urgent need to search for new alternative treatment. PCSK9 is an enzyme responsible in directing LDL-receptor (LDL-R)/LDL-cholesterols (LDL-C) complex to lysosomal degradation, preventing the receptor from recycling back to the surface of liver cells. Therefore, PCSK9 offers a potential target to search for small molecule inhibitors which inhibit the function of this enzyme. In this study, a marine invertebrate *Acanthaster planci*, was used to investigate its potential in inhibiting PCSK9 and lowering the levels of cholesterols. Cytotoxicity activity of *A. planci* on human liver HepG2 cells was carried out using the MTS assay. It was found that methanolic extract and fractions did not exhibit cytotoxicity effect on HepG2 cell line with IC_50_ values of more than 30 µg/mL. A compound deoxythymidine also did not exert any cytotoxicity activity with IC_50_ value of more than 4 µg/mL. Transient transfection and luciferase assay were conducted to determine the effects of *A. planci* on the transcriptional activity of PCSK9 promoter. Methanolic extract and Fraction 2 (EF2) produced the lowest reduction in PCSK9 promoter activity to 70 and 20% of control at 12.5 and 6.25 μg/mL, respectively. In addition, deoxythymidine also decreased PCSK9 promoter activity to the lowest level of 60% control at 3.13 μM. An in vivo study using Sprague Dawley rats demonstrated that 50 and 100 mg/kg of *A. planci* methanolic extract reduced the total cholesterols and LDL-C levels to almost similar levels of untreated controls. The level of serum glutamate oxalate transaminase (SGOT) and serum glutamate pyruvate transaminase (SGPT) showed that the administration of the extract did not produce any toxicity effect and cause any damage to rat liver. The results strongly indicate that *A. planci* produced a significant inhibitory activity on PCSK9 gene expression in HepG2 cells which may be responsible for inducing the uptake of cholesterols by liver, thus, reducing the circulating levels of total cholesterols and LDL-C. Interestingly, A. planci also did show any adverse hepato-cytotoxicity and toxic effects on liver. Thus, this study strongly suggests that *A. planci* has a vast potential to be further developed as a new class of therapeutic agent in lowering the blood cholesterols and reducing the progression of atherosclerosis.

## 1. Introduction

Atherosclerosis is known as a major risk of cardiovascular disease (CVD) [1]. Globally, the prevalent cases of CVD increased to almost double from 271 million in the year 1990 to 523 million cases in 2019. It has been the leading cause of mortality worldwide which increased steadily within the past 20 years, from 12.1 million in 1990 to 18.6 million deaths in 2019 [2].

Atherosclerosis is categorized as an inflammatory disease of the arteries and associated mainly with high concentration of circulating lipids and other metabolic alterations [3]. The formation and progression of atherosclerotic plaque are induced initially by the subendothelial deposition of oxidized low-density lipoprotein-cholesterols (LDL-C) which triggers the uptake of cholesterols by macrophages, which in turn, transformed into lipid-loaded foam cells [4]. These foam cells are then responsible for inducing a cascade of inflammatory responses which include accumulation of additional LDL-C and extracellular matrices which lead to the build-up of plaque inside the walls of arteries [5]. As the arteries hardened and narrowed due to the accumulation and advanced formation of plaques that protrude into the lumen of blood vessels, the blood flow to the vital organs such as heart and brain is impeded which eventually causes the life-threatening CVD such as heart attack and stroke [6].

The most common treatment to prevent the development or to reduce the progression of atherosclerosis is to lower the levels of circulating cholesterols. Currently, the most commonly used drugs for atherosclerosis treatment are statins which inhibit HMG-CoA reductase, a key enzyme in the synthesis of cholesterols in the liver [7]. However, the use of statin may lead to adverse side effects when prescribed to patients in a long term which include liver damage and muscle breakdown with possible damage to the renal tubules [8]. In addition, some of the patients prescribed with statins did not achieve the recommended level although they received a high dosage of statin [9]. 

The discovery of a serine protease protein known as proprotein convertase subtilisin kexin 9 (PCSK9) has paved a way for the development of new therapeutic agents to combat atherosclerosis [10]. PCSK9 is an enzyme present on the liver cell surface together with LDL receptor (LDL-R). Upon the binding of LDL-C, LDL-R is internalized via endocytosis to deliver cholesterols into the cells for further metabolic processes. However, PCSK9 is capable of binding to LDL-R/LDL-C complex at the cell surface and responsible for directing the complex for lysosomal degradation, therefore, preventing the recycling of LDL-R to the cell surface. The reduction in the number of LDL-R leads to a decrease in the uptake of LDL-C by liver cells and an increase in the plasma cholesterols which may trigger the development of atherosclerosis [11]. Therefore, inhibiting PCSK9 increases the availability of LDL-R, which causes an increase in the uptake of LDL-C and clearance from the circulation [12]. Thus, PCSK9 provides a potential target to search for new small molecules that may potentially be developed as therapeutic agents in reducing the progression of atherosclerosis. To date, two monoclonal antibodies (Alirocumab and Evolocumab) and an siRNA-based drug (Inclisiran) have been approved by FDA in 2015 [13] and EMA this year [14], respectively, for clinical use. Unfortunately, although the use of both Alirocumab and Evolocumab may lead to a significant reduction in LDL-C levels and a considerable risk of adverse effects, their inconvenience due to 12 or 24 annual subcutaneous injections and expensive manufacturing costs are a major limitation to practical usage [15]. Similarly, the cost of prescribing inclisiran to patients in a year remains in the similar range as Alirocumab and Evolocumab, and therefore, not economical to prescribe to all patients suffering from hypercholesterolemia as compared to statin, a natural product-based [16].

Various natural products such as berberin, curcumin, resveratrol, epigallocatechin gallate, and welsh onion extract reduced the gene expression of PCSK9, which in turn, decreased the circulating levels of cholesterols (references). However, the study on the effects of marine natural products on PCSK9 gene expression is extremely limited. 

Marine environment is an important source of biological diversity and habitats for invertebrates that provide various structurally-diversed compounds that have vast potential for the discovery of new drugs. The abundance of bioactive compounds have been isolated from marine invertebrates with promising biological activities [17]. Several marine metabolites have been successfully developed as drugs, mostly in the treatment of cancer. For instance, two nucleoside-based compounds, spongothymidine and spongouridine isolated from the Caribbean sponge *Tethya crypta* were used as the basis to synthesise new generation analogues known as vidarabine and cytorabine which have been used clinically as anti-viral and anti-cancer drugs, respectively [18]. 

*Acanthaster planci* which is commonly known as Crown of Thorns from genus Acanthaster and phylum Echinodermata is a venomous starfish that produces compounds such as steroid, alkaloids, peptides, and anthraquinones that exhibit many beneficial biological activities including anti-fungal, anti-viral, cytotoxic, and antimicrobial activities [19,20]. Its advantages were attributed to the presence of asterosaponins, oligoglycosides sulphate, and glycosaminoglycans [21,22,23]. In addition, a compound, methyl benzoate, isolated from *A. planci* increased the gene expression of PCSK9 as well as SR-B1, an important receptor on the liver cells involved in reverse cholesterol transport [24]. Another marine natural product, aaptamine isolated from a marine sponge *Aaptos aaptos*, also produced an inhibitory effect of PCSK9 gene expression [25]. However, none of these studies elucidated the role of these marine natural products in reducing the levels of circulating cholesterols in the in vivo model. 

However, to date, a detailed analysis of the function of natural products prepared from *A. planci* in reducing the plasma cholesterol levels via PCSK9 has not been carried out. Thus, this study was conducted to address those limitations.

## 2. Results

### 2.1. Methanolic Extract and Fractions of A. planci Do Not Produce Cytotoxicity Effects on HepG2 Cell Line

Prior to determining the effects of *A. planci* on PCSK9 promoter activity, the cytotoxicity activity of the natural products on human liver HepG2 cell line was investigated. This step was important to ensure that the samples used to screen for the PCSK9 promoter activity were not cytotoxic on the HepG2 cell line, which was used as the model system in this study. 

As shown in Figure 1, methanolic extract of *A. planci* significantly induced the cell growth when cells were treated at the concentrations between 6.25 and 50 µg/mL. The extract produced the highest growth-inducible activity of 127% when the cells were treated at the lowest concentration of 6.25 µg/mL. The extract did not produce any significant cytotoxicity activity at 100 µg/mL. Interestingly, the cell growth was significantly inhibited to 26% of untreated control at the highest concentration of 200 µg/mL.

The US National Cancer Institute has set a criterium of IC_50_ value above 20 µg/mL for the extract and fraction, and 4 µg/mL for the compound to be categorized as non-cytotoxic [26]. It is also reported that an extract or fraction with IC_50_ value of less than 5 µg/mL was classified as strongly toxic, 5–10 µg/mL toxic, 11–30 µg/mL mildly toxic, and more than 30 µg/mL non-cytotoxic [27]. Therefore, based on the criteria, the extract did not exhibit any cytotoxicity effects on the HepG2 cell line. 

A similar observation was found when the cells were treated with 10 different fractions prepared from methanolic extract. Fractions 1, 3, 4, 5, 9, and 10 did not reduce the cell growth of more than 50% at all concentrations used to treat the cells (Figure 2). Interestingly, Fractions 1, 3, and 5 significantly induced the growth rate at 3.13 µg/mL; 6.25 and 12.5 µg/mL; and 3.13 µg/mL, respectively, as shown in Figure 2. However, Fractions 2, 6, 7, and 8 inhibited the growth lower than 50% of the untreated control only at the highest concentration of 50 µg/mL. These results clearly indicate that the fractions also did not produce any cytotoxicity effect on the HepG2 cell line. 

### 2.2. Methanolic Extract and Fractions of A. planci Reduce the PCSK9 Promoter Activity

The gene-reporter assay was utilised to determine the effects of *A. planci* methanolic extract and fractions on the PCSK9 promoter activity. As shown in Figure 3, there was no significant change in the PCSK9 promoter activity when HepG2 cells were treated with the extract at 3.13 and 6.25 µg/mL. Interestingly, the promoter activity was gradually and significantly reduced at the concentrations of 12.5 µg/mL and above to reach its lowest levels of 67% of control at 50 µg/mL. 

All of the fractions significantly reduced the PSCK9 promoter activity at certain concentrations except Fractions 9 and 10, of which, all of the concentrations used to treat the cells, did not decrease the transcriptional activity of PCSK9 promoter, but, significantly increased the promoter activity which reached the highest level of 178 and 254% of control, at 3.13 µg/mL of Fraction 9, and 25 µg/mL of Fraction 10, respectively. 

Fractions 2, 3, 4, and 6 significantly reduced the PCSK9 promoter at the lowest concentration used in this study (3.13 µg/mL) to 47, 56, 59, and 68% of control, respectively. However, the transcriptional activity was increased again at higher concentrations, reaching similar or higher than that of the untreated control (Figure 4). For example, Fraction 2 at concentrations of 3.13 and 6.25 µg/mL reduced the promoter activity to 47 and 31%, respectively, however, the treatment at higher concentrations of 12.5, 25, and 50 µg/mL increased the activity to reach a similar level as the untreated control at the highest concentration (Figure 4). Similarly, the lowest concentration of Fraction 6 significantly reduced the PCSK9 promoter activity, however, the transcriptional activity was increased higher than that of at 3.13 µg/mL when cells were treated at higher concentrations. Interestingly, the PCSK9 promoter activity was decreased again at the highest concentrations in Fractions 3- and 4-treated cells (at 12.5, 25 and 50, and, 25 and 50 µg/mL, respectively), after it increased to 6.25 µg/mL. 

Fractions 3, 7, and 8 exhibited the lowest level of PCSK9 promoter activity at the highest concentration used in this study to 50, 74, and 70% of control, respectively. Fractions 3 and 8, not only decreased the activity at 50 µg/mL, but also reduced the activity at 3.13 µg/mL, and, 6.25 and 25 µg/mL, respectively.

It is interesting to note also that only Fractions 2 and 5 did not induce the transcriptional activity of PCSK9 promoter higher than that of control. Fraction 2 produced the lowest activity at 6.25 µg/mL to 31% of control, and increased thereafter to a similar level of the untreated control at the highest concentration. Similarly, Fraction 5 significantly reduced the PCSK9 promoter activity to 72% at lower concentration of 6.25 µg/mL and increased gradually at higher concentrations reaching almost to a similar level of untreated control at 12.5 µg/mL.

Overall, only Fractions 2 and 4 decreased the PCSK9 transcriptional activity lower than 50% of control at certain concentrations. Fraction 2 produced the highest inhibitory activity at 6.25 µg/mL of which the PCSK9 promoter activity was reduced to 31% of control, whereby, Fraction 4 exerted the lowest activity at 25 µg/mL with a reduction of PCSK9 promoter activity to 41% of control.

This finding indicates that the *A. planci* methanolic extract and its fractions contained natural products that acted as inhibitors of PCSK9 promoter. 

### 2.3. Deoxythymidine, a Compound Isolated from Fraction 1 Reduces PCSK9 Promoter Activity

Further fractionation of the selected fraction using column chromatography revealed the presence of a major compound. Upon NMR analysis (Figure 5), the compound was identified as deoxythymidine. 

As shown in Figure 6 (A), deoxythymidine did not produce any significant reduction in cell growth indicating that the compound was not cytotoxic on HepG2 cells. In contrast, the cell growth was increased by 18% as compared to the untreated control when the cells were treated with 3.13 µM of the compound. 

Subsequently, deoxythymidine was then used to treat the transfected cells in order to determine its effect on regulating the transcriptional activity of PCSK9 promoter. It was revealed that the PCSK9 promoter activity was significantly reduced to 62 and 69% of control when treated with the compound at 3.13 and 6.25 µM, respectively, then increased to an almost similar level to that of control at 12.5 µM, and significantly decreased again to 75 and 81% of control at 25 and 50 µM, respectively.

### 2.4. Methanol Extract of A. planci Significantly Reduces Plasma Total Cholesterols and LDL-C in Rats

The *A.*
*planci* methanolic extract was demonstrated to decrease the activity of PCSK9 promoter (Figure 3). It was widely demonstrated that PCSK9 plays a pivotal role in the intracellular degradation of cell surface LDL-R, which in turn, decreases the levels of the receptor presence on liver cells and reduces the circulating levels of LDL-C [28]. Therefore, it is vital to investigate the effect of the methanolic extract on the circulating levels of total cholesterol and LDL-C using the in vivo model. In addition, the safety effect of the extract was also investigated by determining the levels of SGPT and SGOT enzymes. 

#### 2.4.1. Analysis of Plasma Total Cholesterol Levels

In order to increase the level of plasma cholesterols, the rats were fed with high cholesterol diet for the first 14 days. The effects of *A. planci* methanolic extract (50 and 100 mg/kg) in reducing the levels of total plasma cholesterols were determined in high fat diet-induced rats. Blood serum was collected and subjected to lipid analysis every 7 days during the experimental period of 42 days. 

As shown in Figure 6, after acclimatisation, the cholesterol levels of rat in all groups on Day 0 in the treatment experiment were in the range between 61.8 and 70.7 mg/dL. In rats that were fed with normal diet throughout the experimental period (Group A), there was a slight decrease of total cholesterol levels from Day 0 until 42. Rats in Groups B, C, D, and E that were fed with high fat diet from Day 0 until 14 showed a significant increase of total plasma cholesterol levels in Day 7 and 14 as compared to control (Day 0). Specifically, the plasma cholesterol levels were increased by 2.2-, 2.3-, 2.0-, and 2.3-fold at Day 7, and, further increased by 2.9-, 2.7-, 2.6-, and 2.5-fold in Groups B, C, D, and E, respectively, at Day 14. 

On Day 14, the diet in Groups B, C, D, and E was switched to a normal diet until Day 42. Rats in Groups C, D, and E were treated with atorvastatin, 100 and 50 mg/kg of the extract, respectively until Day 28, whereas, rats in Group B were left untreated. As shown in Figure 6, the plasma cholesterol levels in Group B continued to increase to 236.3 mg/dL at Day 21 (3.5-fold increase as compared to untreated control) and decreased to 140.4 and 80.1 mg/dL at Day 28 and 35, respectively which were above the control level, and, further reduced to the control level to 68.4 mg/dL at Day 42. However, rats that were treated with atorvastatin showed a marked decrease in the cholesterol levels at Day 21 to 74.8 mg/dL (55% decrease as compared to Day 14), and, to untreated control levels at Day 28, and, even lower than that of control levels at Day 35 and 42 (23.5 and 21.4% decrease as compared to control, respectively). Interestingly, rats that were treated with 50 and 100 mg/kg of methanolic extract of *A. planci* also produced almost a similar pattern of reduction in the plasma cholesterol levels as to rats that were treated with atorvastatin. Specifically, rats in Groups D and E that were treated with 100 and 50 mg/kg of the extract, respectively, showed a reduction in cholesterol levels at Day 21 by 57.9 and 64.0%, respectively, as compared to Day 14. The cholesterol levels remained low and exerted no significant difference as compared to control levels, thereafter until the experimental period ended at Day 42 for both groups.

#### 2.4.2. Analysis of LDL Cholesterol Levels

At Day 0, the plasma level of LDL-C level for all groups was between 27.3 and 32.9 mg/dL. The levels of LDL-C in rats fed with normal diet (Group A) remained stable and no significant change was observed throughout 42 days of the experimental period. Rats in Group B that were fed with high fat diet for 14 days followed by a normal diet from Day 14 onwards exhibited a significant increase in LDL-C level reaching the highest levels of 161.3 ± 32.4 mg/dL at Day 21 (5.0-fold increase as compared to the control). The LDL-C level was then decreased to 113.4 mg/dL (3.5-fold higher than that of control) at Day 28 and approaching the control levels at Day 35 and 42 (significant 1.54- and 1.53-fold higher than that of control). 

Interestingly, in Groups C, D, and E, the LDL-C levels were significantly increased until Day 14 to 120.4 (4.4-fold increase as compared to control), 122.3 (3.7-fold increase), and 134.2 mg/dL (4.4-fold increase), respectively, and drastically reduced at Day 21 to 43.1, 49.1, and 32.4 mg/dL, respectively, and, remained low but not significant as compared to control, thereafter (Figure 7). 

#### 2.4.3. Analysis of Triglycerides Cholesterol Levels

The levels of triglycerides at Day 0 were in the range between 45.44 and 53.11 mg/dL. As expected, the levels of triglycerides in rats fed with normal diet did not produce any significant changes throughout the experimental period. Meanwhile, the levels of triglycerides in rats fed with high fat diet in Groups B, C, D, and E significantly increased at Day 7 and 14 (Figure 8). Interestingly, rats that were fed with atorvastatin (Group C) and 50 and 100 mg/kg of methanolic extract of *A. planci* (Groups D and E, respectively) from Day 14 to 28, produced a significant decrease in the levels of triglycerides at Day 21 and remained low with no significant difference as compared to control levels thereafter. By contrast, the levels of triglycerides in rats of Group B that did not receive any treatment remained significantly higher than that of control levels at Day 21 and 28, before returning to a normal level at Day 35 and 42.

### 2.5. Methanol Extract of A. Planci Does Not Produce Significant Changes in SGOT and SGPT Levels

The measurement of SGOT was used to assess the toxicity of *A. planci* methanolic extract on the liver. As shown in Table 1, the level of SGOT for rats that were fed with normal diet throughout the experimental period (Group A) was within the range of 90.1 and 106.7 U/I. There was no significant difference between the level of the enzyme between all the 7-day interval plasma samples (*p* > 0.05). In Group B, there was no significant difference between the level of SGOT in rats fed with high fat diet (Day 7 and 14) and Day 0, as well as with plasma samples obtained from rats subsequently fed with normal diet at Day 21, 28, 35, and 42 (*p* > 0.05). No significant changes were also observed in SGOT level in plasma samples collected from rats treated with atorvastatin (Day 21 and 28) as well as samples in the recovery phase at Day 35 and 42, as compared to Day 0 (*p* > 0.05). Interestingly, plasma samples collected from rats treated with 100 mg/kg methanolic extract (Group D), showed a reduction at Day 21 and 28 (90.0 and 91.3 U/I, respectively) and further decreased in the recovery phase at Day 35 and 42 (86.3 and 89.3, respectively). However, the reduction in the levels of the enzyme was not significant as compared to control and between samples of 7-day intervals (*p* > 0.05). Similarly, the level of SGOT in Group E also did not produce any significant difference between the enzyme collected from 7-day interval samples (*p* > 0.05).

Since SGPT appears after SGOT as an indicator of the liver damage, the level of this enzyme also acts as the parameter used to assess the toxicity effects on the liver [29]. As shown in Table 2, the SGPT level in rats that were fed with normal diet over the experimental duration of 42 days (Group A) was found between 42.6 and 48.2 U/I with an average of 45.5 U/I. No significant difference was observed between all samples collected from 7-day intervals of 42 days (*p* > 0.05). In Group B, the level of SGPT in high fat diet-fed rats was between 38.6 and 48.7 U/I from Day 0 to 14. The level of the enzyme was also within this range when the diet was changed to normal diet from Day 14 onwards, which was between 41.5 and 48.9 U/I. No significant change was observed between all samples in Group B (*p* > 0.05). The atorvastatin treatment also did not produce any significant difference (*p* > 0.05) in rats (Group D) of which the level of enzyme was between 37.7 and 44.5 U/mL from Day 21 to 42 as compared to between 38.3 and 44.0 U/I from Day 0 to 14. More importantly, the methanolic extract at 100 and 50 mg/kg (Groups D and E, respectively) also did not exert any significant variation in the level of SGPT in all samples collected from rats in 7-day intervals (*p* > 0.05).

## 3. Discussion

A constant elevated level of LDL-C leads to the development of atherosclerosis which in turn, is the main cause of cardiovascular diseases [30]. PCSK9 is an enzyme that plays a pivotal role in the uptake of LDL-C by LDL-R into the liver cells. This enzyme when it binds to the LDL-R/LDL-C complex on the cell surface, directs the complex to lysosomes that not only break down LDL-C into smaller components, but more importantly, destroy LDL-R intracellularly, and thus, prevent the receptor from recycling back to the cell surface. The lack of LDL-R on liver cells leads to a decrease in the uptake of LDL-C which increases the circulating level of LDL-C [31]. Therefore, PCSK9 offers a good and valid target to search for small molecule inhibitors in order to reduce the gene expression of PCSK9. To date, two drugs (Alirocumab and Evolocumab) have been approved by the Food and Drug Administration (FDA) and commercially available for the treatment for hypercholesterolemia since 2015 [32,33]. In addition, another drug (inclisiran) that uses a gene silencing approach that specifically destroys PCSK9 mRNA and inhibits the synthesis of PCSK9 protein has been given an approval by the Committee for Medicinal Products for Human Use (CHMP) of the European Medicines Agency (EMA) to be used in European Union in December 2020 [34,35,36]. However, these drugs are biotechnology-based and the cost of prescribing these drugs to patients are extremely high [15,37], therefore, not economical to prescribe to all patients suffering from hypercholesterolemia as compared to statin, a natural product-based, which is more economical [16]. Thus, natural product-based drugs still offer a cost effective and an economic value for patients in the treatment of various illnesses.

In this study, the effects of marine invertebrate, *Acanthaster planci*, in regulating the gene expression of PCSK9 as well as the plasma levels of total cholesterols and LDL-C were investigated. It is vital to determine the cytotoxicity activity of the extract, fractions and compound isolated from *A. planci* in order to ensure that the natural products would not exert any cell death on human liver cells, HepG2, that were used as the model system in this study. Various studies demonstrated the cell death-inducible effect of marine organisms on HepG2 cells, such as mesohyl of a sponge, *Hemimycle arabica* [38]. In addition, the extract prepared from sea cucumber *Pearsonothuria graeffei*, lollyfish *Holothuria atra*, and sea hare *Aplysia dactylomela* also produced the cytotoxicity activity on HepG2 cell line with IC_50_ of 16.22, 12.48, and 6.51 µg/mL, respectively [39]. In contrast, the extract and fractions of *A. planci* did not produce any cytotoxicity activity on the HepG2 cell line (Figure 1 and Figure 3). A previous study also reported that the methanolic extract of the outer layer and visceral organ of *A. planci* did not induce cell death on HepG2 cells [40]. In addition, the extract and fractions prepared from the whole-body mass of *A. planci* also reported did not exert a cytotoxicity activity on HepG2 cells [41]. Similarly, a compound identified as deoxythymidine that was isolated from *A. planci* did not produce any significant inhibition on the cell growth of HepG2 cells (Figure 5). Moreover, a study from Mat-Lazim et al. demonstrated that a compound isolated from *A. planci**,* methyl benzoate, exhibited non-cytotoxic effects on the HepG2 cell line [24]. The results obtained from this study clearly indicate that the extract and fractions prepared from *A. planci* as well as an isolated compound deoxythymidine, were not cytotoxic on HepG2 cells and considered as safe to be used to determine the effects on PCSK9 gene expression.

As shown in Figure 2 and Figure 4, the extract and fractions of *A. planci* significantly reduced the transcriptional activity of PCSK9 promoter. It was widely demonstrated that various natural products reduced the gene expression of PCSK9. For example, extracts prepared from welsh onion, shepherd’s purse, pigeon pea, and black raspberry produced a significant reduction in PCSK9 gene expression [42,43,44,45]. Interestingly, all these three extracts concomitantly induced the expression of LDL-R as well as the uptake of LDL-C by liver cells. 

It is also interesting to highlight here that natural products prepared from *A. planci* were demonstrated to induce the transcriptional activity of PPAR and SR-B1, which play a pivotal role in lipid metabolism and reverse cholesterol transport [24,40,46]. It further emphasizes the importance of *A. planci* in reducing the levels of blood cholesterols by regulating, not only PCSK9 but other important lipid transport-related genes. 

In addition to the extract and fraction, a compound isolated from *A. planci*, deoxythymidine was also found to reduce the transcriptional activity of PCSK9 promoter (Figure 5). The compound 3′ azido-3′-deoxythymidine also produced other biological activities such as anti-cancer activity as well as anti-HIV properties as it selectively inhibited the reverse transcriptase of the human T-lymphotropic virus by completely blocking the virus replication [47,48,49]. Several compounds isolated from A. *planci* were reported to reduce the gene expression of PCSK9 in HepG2 cells. For instance, methyl benzoate and its derivative, *N*-(2,3-dihydro-1*H*-inden-2-yl)-2-methoxybenzamide inhibited the gene expression of the PCSK9 [25]. Similarly, aaptamine, a compound isolated from other marine organisms, *Aaptos aaptos* was also reported to inhibit the transcriptional activity of PCSK9 [25]. Several other compounds such as berberine, originally isolated from a Chinese herb Huanglian, also reduced the gene expression of PCSK9, and in turn, increased the gene expression of LDLR and uptake of LDL-C by liver cells [50]. In addition, curcumin which is one of the main bioactive polyphenolic components of the spice turmeric, resveratrol originally isolated from the roots of *Veratrum grandiflorum* and also found in red wine, grapes, and peanuts as well as epigallocatechin gallate (EGCG) the most active catechin found in green tea were also shown to exert the hypocholesterolaemic activity by reducing PCSK9 and increasing LDL-R [51,52,53].

It is well known that liver cells uptake LDL-C via the binding of the cholesterols to LDL-R on the cells surface [54]. The ligand-receptor complex is then internalized into the cells via receptor-mediated endocytosis and transported to endosomes where LDL-C is released from LDL-R, of which LDL-C, is then directed to lysosomes to be degraded. LDL-R, in turn, recycles back to the cell surface to further uptake the circulating LDL-C [55]. However, the binding of PCSK9 to the LDL-R/LDL-C complex prevents the release of the cholesterol in endosomes which leads the entire complex to undergo lysosomal degradation, significantly depleting the receptors from the cell surface [56,57,58]. Thus, it is interesting to speculate that the reduction of PCSK9 gene expression by *A. planci* may also lead to an increase in the uptake of LDL-C by liver cells via inducing the level of LDL-R.

Therefore, the results obtained from this study showed the potential of *A. planci* as important resources to search for the candidates that reduce an elevated level of circulating LDL-C. However, the published studies on the effects of marine-based natural products in reducing the cholesterol levels in the in vivo model is extremely limited. Thus, this study was carried out in order to determine the effects of the methanolic extract of *A. planci* in regulating the plasma cholesterols, LDL-C, and triglycerides by utilising rats as the model system.

In order to establish the rat model with high level of cholesterols and LDL-C, rats were fed with high fat diet containing cholesterol (1%) and cholic acid (0.1%) based on previous studies [59,60,61] with modifications for 2 weeks. As shown in Figure 6, the level of total cholesterol was increased from an average of 67.6 mg/dL at Day 0 to 176.2 mg/dL at Day 14 (an increase of 261%) and further increased to 236.3 mg/dL at Day 21 (Group B, without treatment). Similarly, the level of LDL-C was also increased from an average of 30.8 mg/dL at Day 0 to 130.2 mg/dL at Day 14 (an increase of 422%) and to 161.3 at Day 21 in Group B (without treatment). A previous conducted study demonstrated that rats that were fed with high lipid contents were induced to hyperlipidemic stage with an increase of total cholesterols and LDL-C from 89.6 to 184.1 mg/dL, and, from 38.0 to 80.1 mg/dL, respectively [62]. Similarly, a significant increase of total cholesterol levels was also found in Sprague Dawley rats that were fed with high fat diet from 82.2 to 193.2 mg/dL [63]. In addition, it was demonstrated that Sprague Dawley rats fed with cholesterol-enriched food consisting of cholesterol 1% (*w*/*w*), cholic acid 0.1% (*w*/*w*), and vegetable oil 8.5% (*v*/*w*) increased the level of total cholesterol by 132 and 267% from an average value of 65.7 mg/dL after Day 14 and 21, respectively. The average level of LDL-C at 30.0 mg/dL in Day 0 was also increased by 294 and 460% at Day 14 and 21, respectively [61]. It was also reported that hyperlipidemia was marked by the total cholesterol level of more than 200 mg/dL, therefore, based on the level of total cholesterols and LDL-C at Day 21 (Table 2 and Table 3) which were higher than that of the levels obtained by Sa’adah et al. [62], hypercholesterolemia was successfully established in the in vivo model of our study. 

Pangestika et al. also reported that total cholesterol and LDL-C levels of hypercholesterolemic rats started to decrease at Day 21 after treatment with simvastatin and almost reaching back to a normal level at Day 42 [64]. It was also reported that statin also prevented the levels of total cholesterols and LDL-C from increasing towards the hypercholesteraemic conditions in rats fed with high fat diet [63,65]. In addition, statin also decreased the levels of triglycerides in dyslipidemic rats [66]. These findings were inconsistent with the rats treated with atorvastatin in our study of which the levels of total cholesterols and LDL-C were drastically reduced from 165.7 and 120.4 mg/dL at Day 14 to 48.6 and 36.9 mg/dL at Day 42, respectively. 

Our study demonstrated that the *A. planci* methanolic extract at both concentrations of 50 and 100 mg/kg was almost as equally effective in ameliorating plasma cholesterol, LDL-C, and triglycerides levels as atorvastatin (Figure 6, Figure 7 and Figure 8). The high levels of cholesterols, LDL-C and triglycerides in rats were reduced drastically after 1-week treatment with the extract, reaching a normal level thereafter. The study on the effects of natural products prepared from marine invertebrates in reducing the levels of total cholesterols and LDL-C using in vivo models was reported elsewhere although extremely limited. For example, the levels of total cholesterols and LDL-C were reduced by 40% in rats that were administered with Manzamine A isolated from a marine sponge *Acanthostrongylophora ingens* [67]. Other studies demonstrated the actions of marine algae in regulating the lipid profiles in rats. It was reported that extracts prepared from marine brown algae *Echlonia cava* and *Ecklonia stolonifera* decreased the levels of total cholesterols and LDL-C in rats fed with high-fat diet by 22 and 66%, as well as, 20 and 35%, respectively [68,69]. However, those studies did not determine the effects of the extracts on PCSK9 gene expression and therefore, the role of PCSK9 could not be determined. Thus, our study, to our best knowledge, was the first to report the hypocholesterolaemia activity of the extract prepared from marine invertebrate that may be mediated with a decrease in PCSK9 gene expression. 

As previously described, PCSK9 is responsible for regulating the breakdown of LDL-R which are responsible for the uptake of LDL-C [70]. LDL-C, commonly known as “Bad Cholesterol” is one of the primary causes of the atherosclerosis [71]. It is responsible for the build-up of cholesterol in the blood vessel due to the uptake by macrophages that transformed into lipid-loaded foam cells [72]. It was widely reported that a reduction in LDL-C leads to a decrease in the probability of CVD and lowers the lifetime risk of atherosclerotic-associated CVD [73,74]. Thus, it is strongly indicated that *A. planci* caused a reduction in LDL-C by inhibiting the level of PCSK9 and increasing and prolonging the level of LDL-R present on the cell surface of liver. 

Enzymes are excellent markers of tissue damage, as organ or tissue damage causes the release of increased amounts of many enzymes in the bloodstream [75]. Several ranges of safe and normal SGOT and SGPT values in rats were reported previously [76,77]. For instance, the SGOT level in rats was reported within 106 and 125 U/I, respectively. Our study showed that the high fat diet, and, the treatment with atorvastatin and extract of *A. planci* did not exceed the SGOT level as described previously and is still within the safe range of 83 and 115 U/I based on the work carried previously [64]. As for the SGPT level, it was reported that concentrations between 30–60 U/I of the enzyme were considered within the safety range [29]. In addition, it was stated that SGPT values were considered to be toxic and caused damage to the liver function when the enzyme was within the range of 233 to 283 U/I [76,78]. The high fat diet and treatment of atorvastatin and the extract produced the SGPT level within the range of 34.7 and 49.0 U/L in rats in our study, which falls within the normal range as previously described [29,64].

This implied that the administration *A. planc**i* methanolic extract at 100 and 50 mg/kg dosage was still safe for use and does not cause damage to liver functions. This finding is consistent with the study by Jeong et al. who revealed that the utilization of cholesterol dosage of only more than 3% in a high cholesterol diet caused toxicity in the rat liver [79]. 

Overall, the results strongly suggest that the methanolic extract of *A. planci* contained promising small molecule inhibitors that produced a significant effect in decreasing the levels of total cholesterol and LDL-C in rats that were previously fed with high fat diet via PCSK9 inhibition. The *A. planci* methanolic extract was also safe and suitable to be developed further as therapeutic agents for the treatment of hypercholesterolemia.

## 4. Materials and Methods

### 4.1. Sample Collection and Extraction

*Acanthaster planci* was collected from Bidong Island, Terengganu, Malaysia. The organism was identified by Dr. Jasnizat Saidin, a marine biologist, for the species confirmation. The organism was then cut into a small size, the internal organ was removed and freeze-dried in a freeze dryer to remove excess water. The dried sample was ground into a powder form and extracted using successive extraction with hexane followed by methanol. The mixture was filtered and the filtrate collected was subjected to rotary evaporator to yield the thick crude extract. 

### 4.2. Fractionation and Profiling

The *Acanthaster planci* methanolic extract was subjected to thin layer chromatography (TLC) to identity phytochemicals presence in the fractions. TLC was carried out to analyze the components present in the fractions using UV_254nm_ and UV_366nm_ as well as the staining reagent anisaldehyde. 

The methanol extract was subjected to column chromatography (25 × 5 cm) on Si gel 60 230–240 mesh (Merck). The column was packed by hexane and followed by gradient elution of sequential hexane:chloroform (1:1), chloroform, chloroform:ethyl acetate (1:1), ethyl acetate, ethyl acetate:methanol (1:1), and methanol. The fractions (200 mL each) were collected, evaporated to remove excess solvent using a rotary evaporator, and analysed by TLC using CHCl_3_: EtOAc (1:1) as a solvent system. The fractions that showed the same TLC profiling were combined. 

Ethyl acetate fractions were then further subjected to column chromatography (30 × 3 cm) on Si gel 60 230–240 mesh (Merck). The column was packed with chloroform:ethyl acetate (4:6) in 100 mL and eluted gradually with CHCl_3_ and EtOH with the ratio of 4:6; 3:7; 2:8; 0.5:9.5 and 100% EtOH to yield fractions and fractions that produced a similar TLC profiling were combined.

The targeted compound present in the identified fraction was isolated and structurally elucidated. ^1^H- and ^13^C-nuclear magnetic resonance (NMR) was used to determine the number of protons and thirteen-carbons, respectively. Nuclear magnetic resonance (^1^H-NMR) spectra was recorded on Bruker ARX 400 MHz and DMX 600 MHz NMR spectrometer with tetramethysilane (TMS) as an internal standard. The chemical shift was recorded in ppm and multiplicites were reported as singlet, doublet, triplet, and multiplet. The suitable deuterated solvent was used. For more information on the structural elucidation of a compound isolated from *A. planci* EF2 please refer the Appendix A.

### 4.3. Cell Culture

The human hepatocellular carcinoma cell line (HepG2) was maintained in a modified eagle medium (MEM) supplemented with 1% (*v*/*v*) amino acid, 1% (*v*/*v*) sodium pyruvate, 1% (*v*/*v*) antibiotics (penicillin and streptomycin), and 10% (*v*/*v*) fetal bovine serum (FBS) in a humidified incubator containing 5% (*v*/*v*) CO_2_ at 37 °C. The cells were subcultured upon reaching 75% confluency.

### 4.4. Cytotoxicity Assay

The cytotoxicity activity of the *A. planci* methanolic extract, fractions, and compound on HepG2 cells was evaluated using the MTS assay [80]. HepG2 cells were seeded onto 96-well plates at 8000 cells/well and grown at 37 °C in a humidified incubator supplemented with 5% (*v*/*v*) CO_2_ for 24 h. Subsequently, cells were treated with methanolic extract and fractions at 50, 25, 12.5, 6.25, and 3.13 μg/mL, and, compound at 3.13, 6.25, 12.5, 25, and 50 μM. Cells were also treated with 1% (*v*/*v*) dimethylsulphoxide (DMSO) as negative control, while the positive control cells were treated with vincristine sulphate [81]. Cells were then incubated for 72 h at 37 °C in 5% (*v*/*v*) CO_2_ incubator. After incubation, cells were assayed using the CellTiter 96 Aqueous one Solution Cell Proliferation Assay System (Promega, Madison, WI, USA). In addition, 20 μL [3-(4,5-dimethylthiazol-2yl)-5(3-carboxymethoxyphenyl)-2-(4-sulfopgenyl)-2H-tetrazolium (MTS) solution was added into each well and incubated for 3.5 h in the humidified 5% (*v*/*v*) CO_2_ incubator at 37 °C. Wells with complete medium and MTS solution without cells were used as the blank. Absorbance was determined at 490 nm using the Glomax Multi Detection System (Promega). 

### 4.5. Transient Transfection and Luciferase Assay

Before transfection, HepG2 cells were seeded onto 96-well plates at 40,000 cells/well and cultured for 24 h in serum-free medium OPTIMEM. Transient transfection was carried out using Lipofectamine LTX with PLUS reagents (Invitrogen, Waltham, MA, USA) according to the manufacturer’s recommendations. The cells were transiently transfected with PGL3 plasmid (Promega) harbouring the firefly luciferase reporter gene linked to the PCSK9 promoter [82]. As an internal control for transfection efficiency, the cells were cotransfected with a *Renilla* luciferase-encoding reference plasmid, pRL-tk (Promega). After 5 h of incubation, the transfected cells were treated with the extract, fractions, and compound as described in the cytotoxicity assay. Cells treated with 1% (*v*/*v*) DMSO were designated as negative control, while positive control cells were treated with berberine sulphate at 20 µM [83]. The treated cells were incubated for 24 h at 37 °C in a humidified incubator supplemented with 5% (*v*/*v*) CO_2_.

The luciferase assay was carried using the Dual Glo Luciferase Assay System (Promega) with modifications [84,85]. Briefly, 90 µL of Dual-Glo^®^ Luciferase Reagent was added into each well and incubated for 10 min. Subsequently, the firefly luminescence signal was measured with Glomax Multidetection System. Then, 90 µL of Dual Glo^®^ Stop & Glo^®^ Reagent was added into each well, incubated for 10 min, and *Renilla* luminescence signal was measured with Glomax Multidetection System. Firefly luciferase absorbance reading was then normalised against *Renilla* luciferase activity. 

### 4.6. In Vivo Study

In this study, 40 male Sprague Dawley rats (species of *Rattus norvegicus*) weighing 100–200 g were used. All the rats were clinically healthy without any disease. The animals were kept in the animal room at Institute of Marine Biotechnology (IMB), Universiti Malaysia Terengganu (UMT). The condition of animal house that used to keep rats were in the standard environment condition and rats were fed with commercial pellet and water made available ad libitum, and, observed for any changes or abnormality during a 7-day acclimatisation. 

Rats were randomly assigned into one group of normal (designated as Group A) and four groups of high fat diet (designated as Groups B, C, D, and E). Each group contained eight rats. The normal diet consisted of standard commercial pellet and the high fat diet was composed of normal pellet supplemented with cholesterol (1%) and cholic acid (0.1%) [58,59,60]. Group A was fed with normal diet from Day 0 to 28 and Groups B, C, D, and E were given high fat diet from Day 0 to 14 (fat-induced period), followed by normal diet from Day 15 to 28. Groups C, D, and E were treated with Atorvastatin (10 mg/kg), methanolic extract at 100 and 50 mg/kg, respectively, once daily by oral administration from Day 15 to 28 known as the treatment period. Subsequently, all the rats were fed with normal diet without any drug or extract treatment from Day 29 to 42 which was known as a recovery period. Overall, the in vivo experimental design was shown in Table 3. Blood was withdrawn from tail vein every week over an experimental period of 42 days for the analysis of plasma total cholesterol (TC) and LDL-C, as well as the level of SGOT and SGPT [86]. 

#### 4.6.1. Total Cholesterol ASSAY

This analysis was carried out to determine total cholesterol levels in the blood of rats from all groups. The analysis was conducted using an enzymatic-colourimetric method, in which the blood plasma samples were analysed by BP Clinical Lab (Terengganu, Malaysia) using the standard protocol.

#### 4.6.2. LDL Cholesterol Assay

LDL cholesterol levels in the blood from rats of all groups were determined using an enzymatic method, in which the blood plasma samples were analysed by BP Clinical Lab (Commercial lab analysis, Terengganu, Malaysia) using the standard protocol.

#### 4.6.3. Triglycerides Assay

Blood was withdrawn from tail vein every week over an experimental period of 42 days for analysis of the triglyceride (TG) level. Then, the samples were processed and analysed using the standard protocol of enzymatic method by BP Clinical Lab (Terengganu, Malaysia).

### 4.7. Statistical Analysis 

All in vitro data were carried out in replicates as indicated in the respective figures and tables, and, expressed as mean ± SD (standard deviation). While all in vivo data were analysed by SPSS 20. The determination of significant differences for all analysed parameters was performed by one way ANOVA (analysis of variance) and followed by Duncan Test with a significant level at *p* < 0.05.

## 5. Conclusions

The methanolic crude extract, fractions, and a compound, deoxythymidine, isolated from a marine invertebrate, *A. planci*, demonstrated a good activity as PCSK9 inhibitors. In addition, the in vivo study also revealed that the methanolic extract decreased the levels of total cholesterols and LDL-C in rats fed with high fat diet. Plasma levels of SGOT and SGPT were observed in the normal range in rats that were treated with the extract which indicates that it did not introduce any toxicity effect to the liver. The findings concluded that *A. planci* contained promising PCSK9 small molecule inhibitors that have vast potential to be developed as therapeutic agents against hypercholesterolemia which in turn, will reduce the progression or prevent the development of atherosclerosis and cardiovascular diseases.

## Figures and Tables

**Figure 1 molecules-26-05094-f001:**
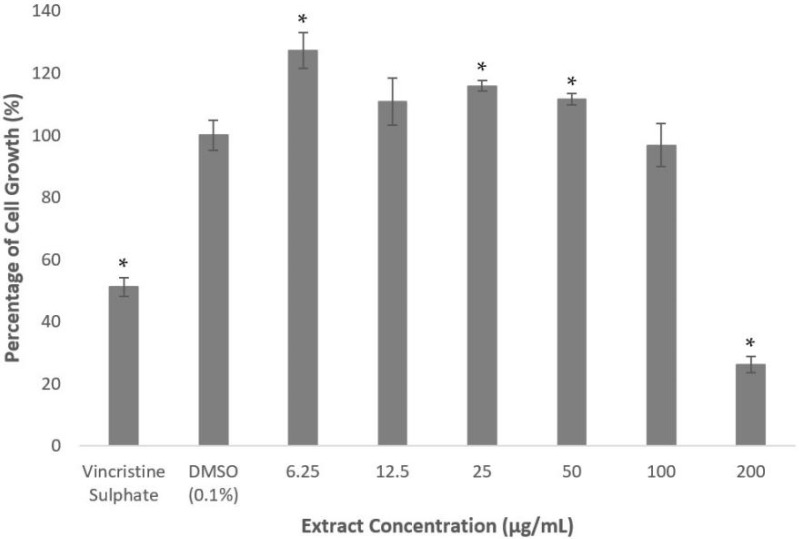
Percentage of cell growth of HepG2 cell line after the treatment of *A. planci* methanolic extract for 72 h. The cells were treated at various concentrations of the extract from 6.25 to 200 µg/mL. The value in the untreated control was assigned as 100% and the values in the treated samples were relative to the untreated control value. Data presented as mean ± standard deviation (SD) with *n* = 6. * denotes significantly different as compared to DMSO-treated cells (negative control) at *p* < 0.05. DMSO was used as the carrier to dissolve and dilute the extract.

**Figure 2 molecules-26-05094-f002:**
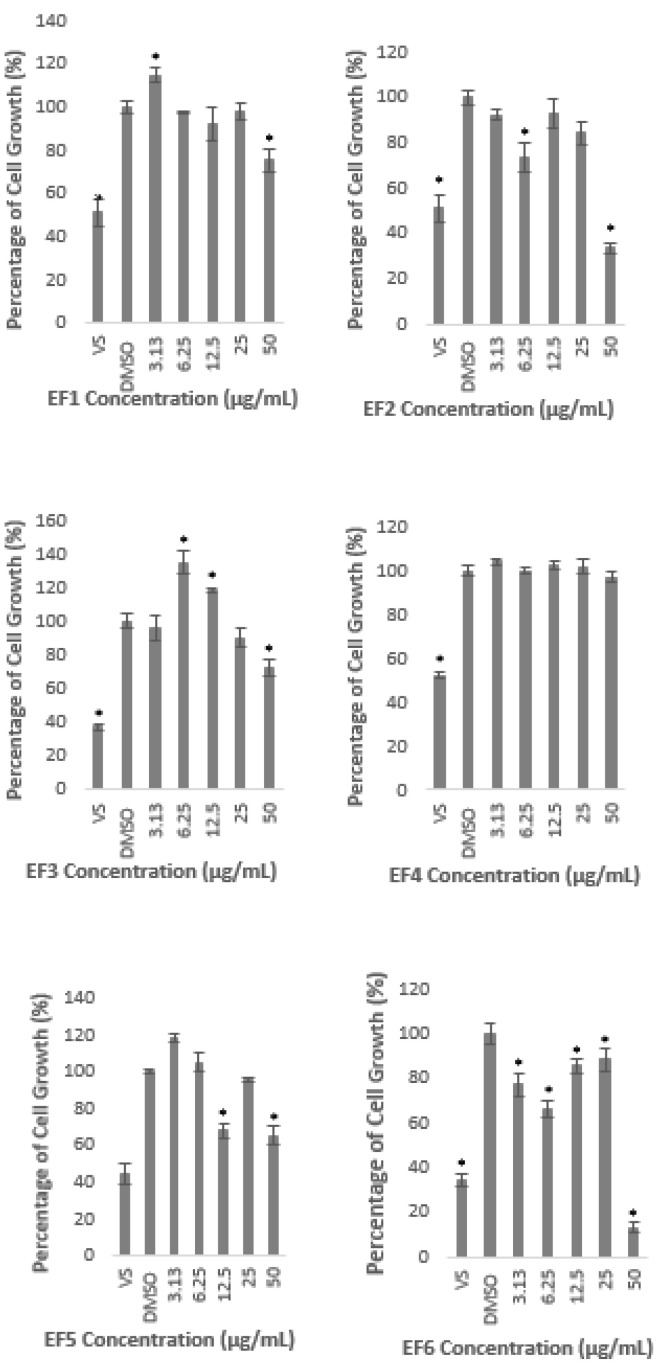
Percentage of cell growth of HepG2 after the treatment with *A. planci* fractions at five different concentrations from 3.13 to 50 µg/mL for 72 h. The percentage of cell growth was compared to the negative control of 1% (*v*/*v*) of DMSO which was used as the carrier. Data obtained presented as mean ± SD with *n* = 6. EF represents the fraction number. * denotes significantly different as compared to DMSO-treated cells (negative control) at *p* < 0.05. DMSO was used as the carrier to dissolve and dilute the extract.

**Figure 3 molecules-26-05094-f003:**
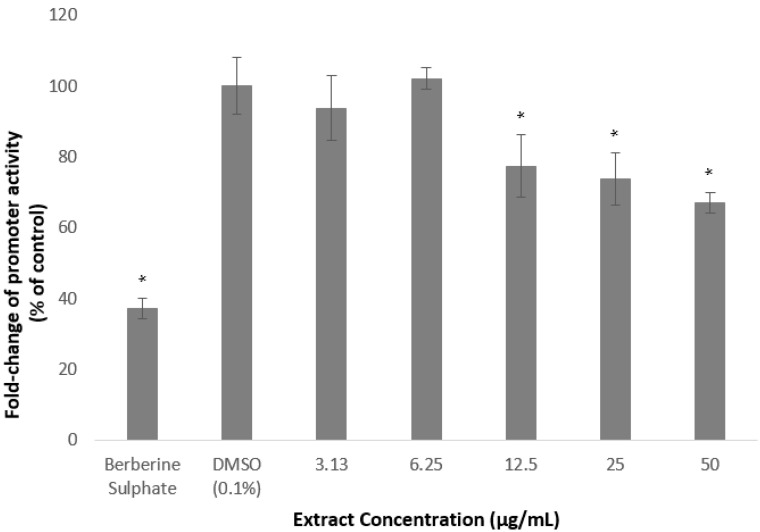
Luciferase activity of HepG2 cells transfected with PCSK9 promoter-reporter construct after 24 h treatment with *A. planci* methanolic extract. The cells were treated at various concentrations from 3.13 to 50 µg/mL. Data presented as mean ± SD with *n* = 6. The value at each point represents the fold change of normalised PCSK9 promoter activity relative to the untreated control which was assigned as 100%. * denotes significantly different as compared to DMSO-treated cells (negative control) at *p* < 0.05. DMSO was used as the carrier to dissolve and dilute the extract.

**Figure 4 molecules-26-05094-f004:**
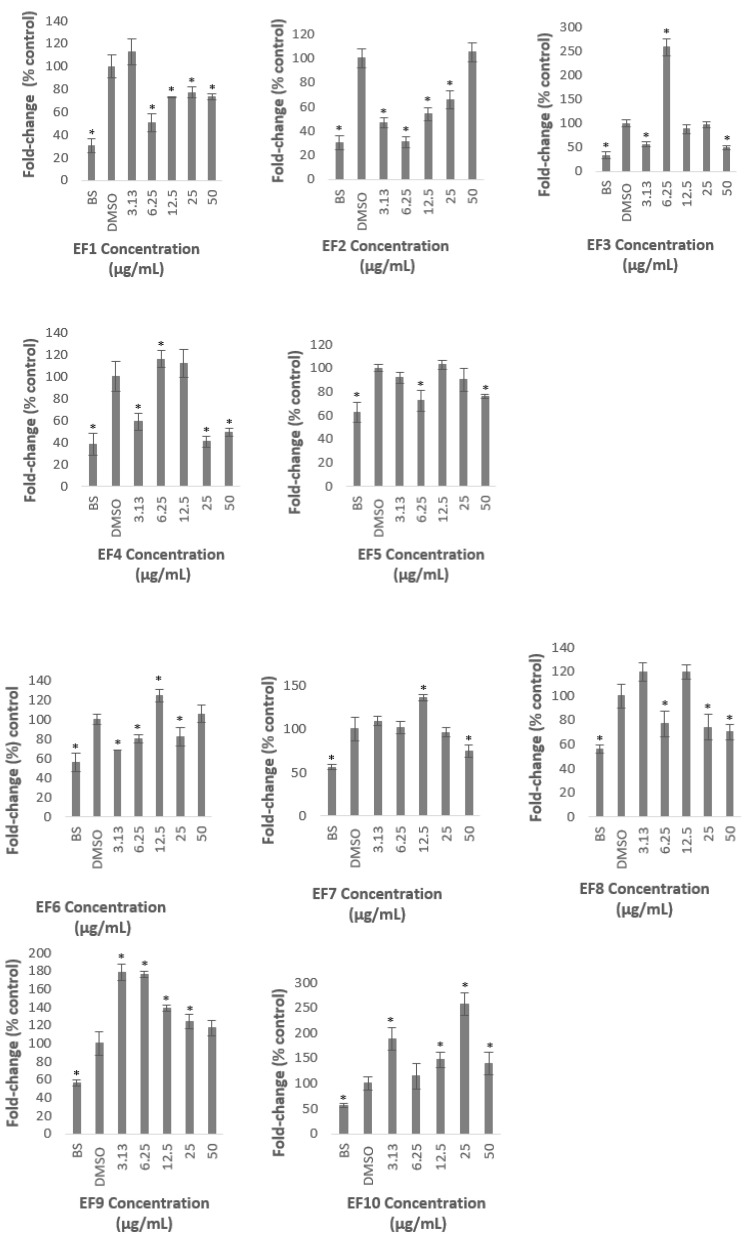
Luciferase activity of HepG2 cells transfected with PCSK9 promoter-reporter construct after 24 h treatment with *A. planci* fractions. The treatments were at five different concentrations from 3.13 to 50 µg/mL. Data presented as mean ± SD with *n* = 6. The value at each point represents the fold change of normalised PCSK9 promoter activity relative to the untreated control which is assigned as 100%. * denotes significantly different as compared to DMSO-treated cells (negative control) at *p* < 0.05. DMSO was used as the carrier to dissolve and dilute the extract.

**Figure 5 molecules-26-05094-f005:**
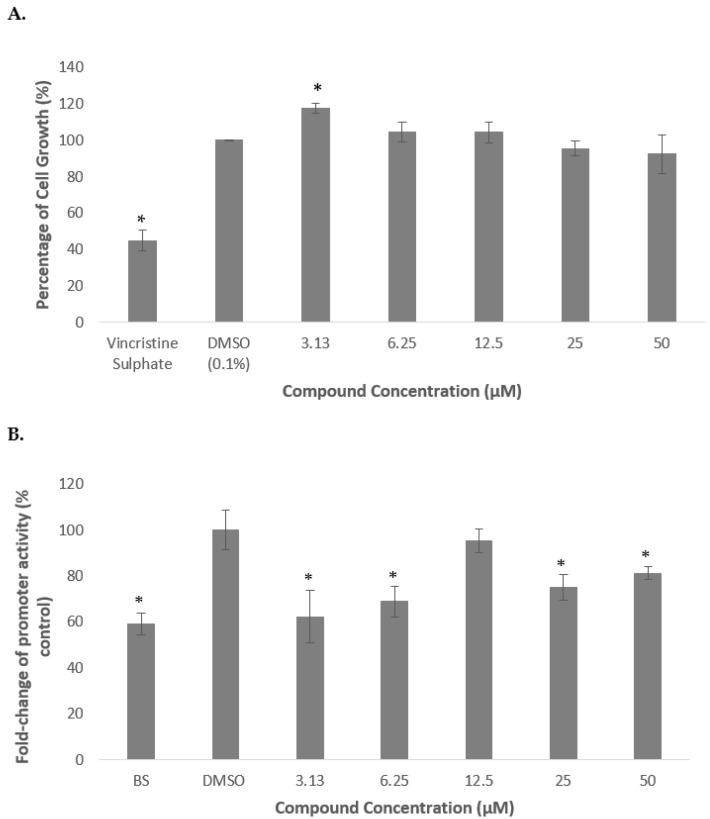
Percentage of cell growth of HepG2 cells after the treatment with deoxythymidine for 72 h (**A**). Luciferase activity of HepG2 transfected with PCSK9 promoter-reporter construct after 24 h treatment with deoxythymidine (**B**). The cells were treated at various concentrations from 3.13 to 50 µM. Data presented as mean ± SD with *n* = 6. The values were compared to the negative control. * denotes significantly different as compared to DMSO-treated cells (negative control) at *p* < 0.05. DMSO was used as the carrier to dissolve and dilute the extract.

**Figure 6 molecules-26-05094-f006:**
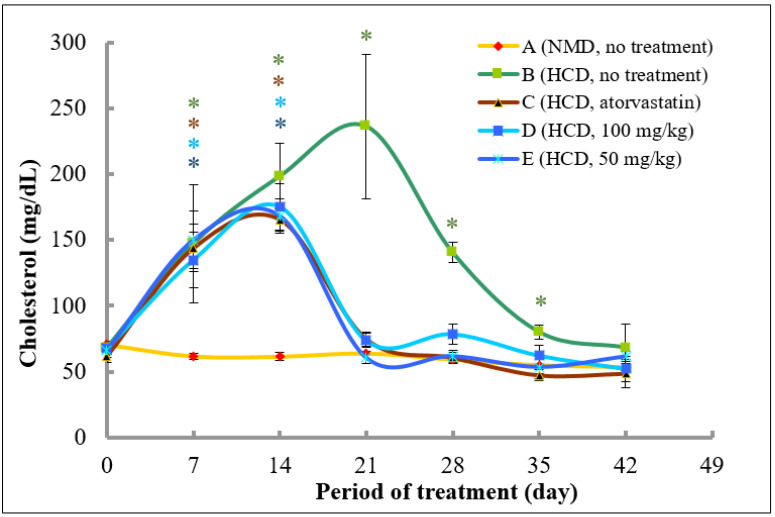
The changes of total cholesterol levels in rats fed with normal diet throughout 42 days of the experimental period. (A) Fed with normal diet for 14 days, followed by high fat diet for another 14 days, followed by normal diet until Day 42 (B); fed with high fat diet for 14 days, followed by normal diet together with atorvastatin for another 14 days, followed by normal diet without atorvastatin until Day 42 (C); fed with high fat diet for 14 days, followed by normal diet together with 100 mg/kg of methanolic extract for another 14 days, followed by normal diet without the extract until Day 42 (D); and fed with high fat diet for 14 days, followed by normal diet together with 50 mg/kg of methanolic extract for another 14 days, followed by normal diet without the extract until Day 42 (E). * denotes significantly different as compared to untreated control rats at Day 0 (negative control) of the respective group at *p* < 0.05.

**Figure 7 molecules-26-05094-f007:**
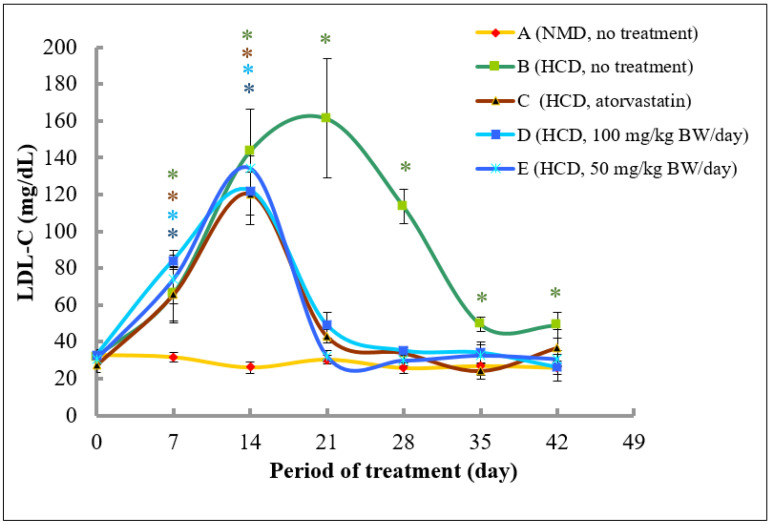
The changes of LDL-C (mg/dL) level in rats fed with normal diet throughout 42 days of the experimental period. (A) Fed with normal diet for 14 days, followed by high fat diet for another 14 days, followed by normal diet until Day 42 (B); fed with high fat diet for 14 days, followed by normal diet together with atorvastatin for another 14 days, followed by normal diet without atorvastatin until Day 42 (C); fed with high fat diet for 14 days, followed by normal diet together with 100 mg/kg of methanolic extract for another 14 days, followed by normal diet without the extract until Day 42 (D); and fed with high fat diet for 14 days, followed by normal diet together with 50 mg/kg of methanolic extract for another 14 days, followed by normal diet without the extract until Day 42 (E). * denotes significantly different as compared to untreated control rats at Day 0 (negative control) of the respective group at *p* < 0.05.

**Figure 8 molecules-26-05094-f008:**
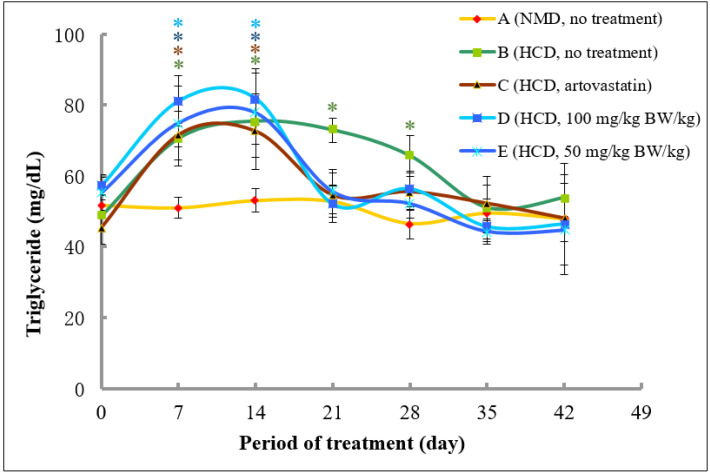
The changes of triglyceride (TG) levels in rats fed with normal diet throughout 42 days of the experimental period. (A) Fed with normal diet for 14 days, followed by high fat diet for another 14 days, followed by normal diet until Day 42 (B); fed with high fat diet for 14 days, followed by normal diet together with atorvastatin for another 14 days, followed by normal diet without atorvastatin until Day 42 (C); fed with high fat diet for 14 days, followed by normal diet together with 100 mg/kg of methanolic extract for another 14 days, followed by normal diet without the extract until Day 42 (D); and fed with high fat diet for 14 days, followed by normal diet together with 50 mg/kg of methanolic extract for another 14 days, followed by normal diet without the extract until Day 42 (E). * denotes significantly different as compared to untreated control rats at Day 0 (negative control) of the respective group at *p* < 0.05.

**Table 1 molecules-26-05094-t001:** Level of plasma SGOT in rats fed with normal diet throughout 42 days of the experimental period. (A) Fed with normal diet for 14 days, followed by high fat diet for another 14 days, followed by normal diet until Day 42 (B); fed with high fat diet for 14 days, followed by normal diet together with atorvastatin for another 14 days, followed by normal diet without atorvastatin until Day 42 (C); fed with high fat diet for 14 days, followed by normal diet together with 100 mg/kg of methanolic extract for another 14 days, followed by normal diet without the extract until Day 42 (D); and fed with high fat diet for 14 days, followed by normal diet together with 50 mg/kg of methanolic extract for another 14 days, followed by normal diet without the extract until Day 42.

Group	The Level of SGOT (U/I) during the Treatment Period Measured Every 7 Days						
	Day 0	Day 7	Day 14	Day 21	Day 28	Day 35	Day 42
A	102.0 ± 19.8	90.1 ± 10.2	91.7 ± 3.8	96.3 ± 12.4	106.7 ± 15.9	94.3 ± 15.9	97.8 ± 10.2
B	103.7 ± 9.8	101.6 ± 15.7	99.3 ± 17.5	100.8 ± 7.9	102.1 ± 18.3	99.0 ± 5.4	114.3 ± 10.2
C	97.5 ± 13.2	104.0 ± 14.7	96.5 ± 16.5	102.5 ± 11.1	106.6 ± 11.1	108.7 ± 11.4	104.0 ± 24.5
D	105 ± 2.4	98.0 ± 10.1	96.8 ± 15.0	90.0 ± 5.0	91.3 ± 3.6	86.3 ± 2.0	89.3 ± 5.2
E	92.5 ± 11.5	86.6 ± 4.2	89.6 ± 6.0	95.4 ± 13.3	100.0 ± 17.2	97.0 ± 8.3	91.0 ± 2.2

Note: Data presented as mean ± SD.

**Table 2 molecules-26-05094-t002:** Levels of plasma SGPT in rats fed with normal diet throughout 42 days of the experimental period. (A) Fed with normal diet for 14 days, followed by high fat diet for another 14 days, followed by normal diet until Day 42 (B); fed with high fat diet for 14 days, followed by normal diet together with atorvastatin for another 14 days, followed by normal diet without atorvastatin until Day 42 (C); fed with high fat diet for 14 days, followed by normal diet together with 100 mg/kg of methanolic extract for another 14 days, followed by normal diet without the extract until Day 42 (D); and fed with high fat diet for 14 days, followed by normal diet together with 50 mg/kg of methanolic extract for another 14 days, followed by normal diet without the extract until Day 42.

Group	The Level of SGPT (U/I) during the Treatment Period Measured Every 7 Days						
	Day 0	Day 7	Day 14	Day 21	Day 28	Day 35	Day 42
A	42.6 ± 7.8	46.1 ± 8.0	46.9 ± 7.2	48.2 ± 3.3	44.7 ± 6.3	46.5 ± 7.6	43.8 ± 11.9
B	42.4 ± 4.7	38.6 ± 7.1	48.7 ± 6.2	48.9 ± 7.2	45.2 ± 6.9	42.7 ± 8.4	41.5 ± 17.1
C	39.5 ± 4.4	38.3 ± 5.8	44.0 ± 3.3	44.5 ± 7.2	43.3 ± 2.4	38.0 ± 2.9	37.7 ± 4.6
D	40.6 ± 7.1	44.2 ± 4.8	41.0 ± 6.3	40.3 ± 5.0	43.8 ± 4.5	49.0 ± 7.1	45.0 ± 8.5
E	42.1 ± 6.7	45.8 ± 5.6	43.0 ± 4.1	42.5 ± 7.0	43.8 ± 6.2	34.7 ± 8.1	42.3 ± 7.6

Note: Data presented as mean ± SD.

**Table 3 molecules-26-05094-t003:** Rats grouping for the in vivo study with 28 days treatments (*n* = 8).

Group	Number of Rats	Diet	Treatment
A	8	Normal diet for 42 days	No treatment
B	8	High fat diet for 14 days then continued with normal diet for an additional 28 days	No treatment
C	8	High cholesterol diet for 14 days then continued with normal diet for an additional 28 days	Atorvastatin (positive control) from Day 15 to 28
D	8	High cholesterol diet for 14 days then continued with normal diet for an additional 28 days	*A. planci* methanolic extract dose 1 (100 mg/kg) from Day 15 to 28
E	8	High cholesterol diet for 14 days then continued with normal diet for an additional 28 days	*A. planci* methanolic extract dose 2 (50 mg/kg) from Day 15 to 28

## Data Availability

All data analyzed during this study are included in this article.

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
