# Peer review of "Acanthaster planci* Inhibits PCSK9 and Lowers Cholesterol Levels in Rats"

_molecules, 2021, doi:10.3390/molecules26165094_

Round 1

Reviewer 1 Report

This study on the beneficial effects of Acanthaster planci extracts on atherosclerosis, through the inhibition of the PCSK9 enzyme, is of a remarkable level. The manuscript is complete and also easy to read for non-expert readers, excellent iconographic presentation, certainly of considerable interest in the medical field. The only remark concerns the statistics.

In light of this, I recommend publication after a simple statistic revision.

Figs 1-5: the data are reported as a percentage, in this case it is not correct to use the ANOVA test, it is necessary to use a non-parametric one, for example Krusal-Wallis.

Tables 1 and 2: add after “The level of SGOT” and “The level of SGPT” the measure used

Reviewer 2 Report

This manuscript investigates new potential therapeutics for treating atherosclerosis. Given the prevalence of atherosclerosis and cardiovascular disease globally, this is an important enterprise. However, I believe there are several things that should be improved before this manuscript should be published:

  1. In the introduction the authors mention several therapeutic agents that can be used to reduce the progression of atherosclerosis. I believe it would be helpful to expand this section to discuss the benefits and limitations of previously developed therapeutics. This would more clearly indicate the significance of the authors' search for additional therapeutics. 
  2. In figure 2, the positive control (vincristine sulphate) produces significantly lower cell growth for fractions 7 and 8 than all of the other fractions (~20% compared to 40-60% for all other fractions). Do the authors have any explanation for why this is the case? This variation in growth between samples seems a bit concerning.
  3. Figure 3 and 4 show that the A. planci extract reduces the activity of the PCSK9 reporter. However, I wonder if the authors have any data on the specificity of the extract? If the A. planci extract is likely to be a useful therapeutic agent, it is important to know if there are any other targets of the extract.
  4. Were any other compounds isolated from the fractions besides deoxythymidine? Deoxythymidine is not a particularly novel compound, as deoxythymidine and derivatives have been used to treat several diseases. The significance of this study would be increased if other isolated compounds were also identified.
  5. Why were the cholesterol levels only measured with the methanol extract and not individual fractions or isolated compounds (such as deoxythymidine)? If the goal is to identify specific therapeutic agents for treating atherosclerosis, I believe it would be more impactful to identify specific compounds that lower cholesterol levels.
  6. Some sections of the discussion might be better moved to the introduction. For example, the paragraphs that begin on lines 210 and 292 contain material that would be very useful and relevant in the introduction.
  7. In line 353, the authors are missing the MHz units after each of the NMR instruments used.
  8. Line 412 refers to atorvastation. I believe this should be atorvastatin.
  9. A figure providing the details of the compound identification could be useful to include as supplementary material.

Reviewer 3 Report

This paper introduces the methanolic crude extract, fractions and a compound, deoxythymidine, isolated from a marine invertebrate, A.planci, which demonstrated a good activity as PCSK9 inhibitors. In addition, the methanolic extract decreased the levels of total cholesterols and LDL-C in vivo study. The results provide A.planci contained promising PCSK9 small molecule inhibitors that have vast potential to be developed as therapeutic agents against hypercholesterolemia. This research may provide new materials for reducing the progression of atherosclerosis and cardiovascular diseases or preventing their development. In general, experiments in this study were designed in reason and properly conducted. The presentation of results is clear and conclusions are appropriate and supported by these results. However, there are many detailed errors in the writing and typesetting of the article. I would recommend this work for publication after major revisions and additions.

Other comments:

  1. The abstract should be a high-level summary of work. However, the authors used a lot of text in the abstract to introduce data of the experimental results, which made some conclusions not clearly displayed. The abstract can be more refined, while clearly showing the conclusions and highlights of this research.
  2. The format of the text in Table 2 is inconsistent with Table 1, obviously it has not been reset.
  3. Figure7 and Table3: there is a problem with typesetting, that some fonts overlap. And the title of the Y axis is not consistent with the article description in Figure 7.
  4. Line 161: “PCSK9 in an enzyme” should be revised to “PCSK9 is an enzyme”.
  5. Figure 4: The results of some fractions at five different concentrations are irregular. When the concentration is 50 µg/mL, the results are less reliable than the previous concentrations.
  6. Line 431: The line spacing of the subtitle is incorrect.
  7. The authors confirmed the methanolic extract could decrease the levels of total cholesterols and LDL-C in rats fed with high fat diet. So have you tested the levels of TG and TC, which are also indicators against hypercholesterolemia? At the same time, you can also test the effect of its combined use with atorvastatin.

Round 2

Reviewer 2 Report

The authors have addressed my criticisms, but did not add any additional experiments that could have improved the impact of the manuscript. While I believe the manuscript is scientifically sound, I think the impact is not as high as it could be since they did not identify any novel compounds from the A Planci organism. With that said, I believe this manuscript is worth publishing as it does show that A. Planci extracts have potential as treatment for cardiovascular disease.

Reviewer 3 Report

The authors sufficiently provided additional data and comments that were suggested by reviewer.